# Beneficial Effects of Milk-Derived Extracellular Vesicles on Liver Fibrosis Progression by Inhibiting Hepatic Stellate Cell Activation

**DOI:** 10.3390/nu14194049

**Published:** 2022-09-29

**Authors:** Shimon Reif, Ariel Atias, Mirit Musseri, Nickolay Koroukhov, Regina Golan Gerstl

**Affiliations:** Department of Pediatrics, Hadassah-Hebrew University Medical Center, Jerusalem 9112001, Israel

**Keywords:** hepatic stellate cells, liver fibrosis, extracellular vesicles, milk, miRNA

## Abstract

Liver fibrosis is the consequence of various chronic liver diseases, resulting in accumulation of extracellular matrix, following the activation and proliferation of hepatic stellate cells (HSCs). Based on the milk-derived extracellular vesicles’ (MDEs’) characteristics and biological proprieties, we investigate whether MDEs may regulate fibrotic progression by inhibiting HSCs’ activation via the MDEs’ miRNA content. In order to study this question, we examined the effect of human and cow MDEs on HSCs isolated from murine livers, on activation, proliferation and their proteins’ expression. We have shown that MDEs are able to enter into HSCs in vitro and into the livers in vivo. MDEs inhibited HSCs’ proliferation following stimulation with PDGF. Moreover, in vivo treatment with MDEs resulted in an increase of in miRNA-148 and Let7a expression in HSCs. In contrast, treatment with MDEs reduced the expression of miR-21 in HSCs. In addition, MDEs regulate HSC activation, as was shown by downregulation of collagen I expression and alpha smooth muscle actin, and upregulation of PPARγ. MDEs carrying beneficial miRNAs can be a nontoxic natural target for treatment of liver cirrhosis.

## 1. Introduction

Liver fibrosis accounts for over 1 million deaths per year worldwide, and is a significant public health concern [1]. Liver fibrosis is the result of chronic, and sometimes acute, damage to the liver, and the excessive accumulation of extracellular matrix (ECM) proteins, which leads to the destruction of liver architecture, and liver cell dysfunction [2].

Advanced liver fibrosis results in cirrhosis, liver failure, and portal hypertension, and often requires liver transplantation [3]. Fibrosis is a conserved response to hepatic injury, occurring in almost all types of diseases with hepatocellular death. The development of liver fibrosis is observed in patients with viral hepatitis, nonalcoholic fatty liver disease (NAFLD), alcoholic liver disease (ALD), cholestatic liver disease, and autoimmune liver disease [4].

Hepatic stellate cells (HSCs) have been identified as the main ECM proteins producing cells in the injured liver. HSCs are the major storage sites of vitamin A in the normal liver. Following chronic liver injury, damaged hepatocytes release free radicals, transforming growth factor-β (TGF-β1) and other fibrogenic and inflammatory mediators that induce the recruitment of inflammatory cells and the activation of HSCs, resulting in increased cellular proliferation and biotransformation from a quiescent vitamin A-storing cell to an activated myofibroblast-like cell. Inflammatory cells stimulate ECM protein synthesis and secretion in the activated HSCs, predominantly collagen type I, leading to liver fibrosis [3].

Extracellular vesicles (EVs) have recently emerged as a potential new treatment modality of liver fibrosis [5]. EVs are nanovesicles released to the extracellular space and transport microRNA (miRNA), mRNA, and proteins. They can transfer their cargo to recipient cells, thus serving as extracellular messengers, to mediate cell–cell communication [3]. MiRNAs regulate a wide range of cellular functions, such as cell differentiation, proliferation and apoptosis [6]. It is thought that miRNA expression may play an important role in the development and prevention of diseases, such as cancer [6], diabetes [7], and hepatic fibrosis [8,9]. Previous studies show that miRNAs play both profibrogenic and antifibrogenic roles in the process of liver fibrosis [8], by regulating the TGF-β1 pathway and targeting SMAD proteins in the liver [10,11].

EVs have been found in different physiological fluids, such as bronchoalveolar lavage fluid [12], blood [13], urine [14], and breast milk (in a higher concentration) [15]. Milk-derived EVs (MDEs) have been found to protect miRNAs and proteins from degradation in the digestive tract. They can thus transfer them into the intestine and facilitate their uptake and endocytosis [16].

MDEs carries beneficial, immune-related miRNAs, such as miR-148, representatives of the Let7 family, miR-26, miR-320, miR-21, and miR-375 [17]. Furthermore, MDE modifies target gene expression, such as DNA methyltransferase (DNMT) [17], and promotes proliferation and differentiation of intestinal epithelial cells [18,19]. Moreover, MDEs have a therapeutic and anti-inflammatory effect in DSS-induced colitis, involving several complementary pathways in its mechanism of action [20].

Based on the MDEs’ characteristics and biological proprieties, we investigate whether MDEs may regulate fibrotic progression by inhibiting HSCs’ activation, via their miRNA content. In order to study this question, we study the effect of human and cow MDEs on HSCs isolated from murine livers. Our results show that MDEs regulate proliferation and activation of HSCs. Hence, the results of this study might provide a new approach for the treatment of liver fibrosis.

## 2. Materials and Methods

### 2.1. Milk Sample Collection

Milk samples were collected from healthy mothers during the first 60 days after delivery. Cows’ milk was collected before pasteurization, from a pool of milk. Samples were transported to the laboratory and stored at −80 °C, until extracellular vesicles’ (EVs) isolation was undertaken.

### 2.2. Extracellular Vesicles’ Isolation from Milk

Extracellular vesicles were isolated by sequential ultracentrifugation and filtration. The milk samples were fractionated by centrifugation at 3000× *g* for 30 min at 4 °C. Two fractions were obtained from each sample: fat and skim milk. The EVs were isolated from the skim fraction. The skim milk was centrifuged at 12,000× *g* for 1 h at 4 °C to remove debris. The skim was then passed through 0.45 μm and 0.22 μm filters to remove residual debris. The filtered supernatant was centrifugated at 135,000× *g* for 90 min at 4 °C to pellet the EVs. The EVs pellet was left over night in phosphate-buffered saline (PBS) at 4 °C to dissolve the exosomes. Milk-derived extracellular vesicles (MDEs) were filtered (0.22 μm). The protein content of the EVs’ preparation was measured by a bicinchoninic acid (BCA) protein assay.

### 2.3. Nanoparticle Analysis

Nanoparticle tracking analysis was performed using an NS300 nanoparticle analyzer (NanoSight, Malvern, Worcestershire, UK), which was used to measure the size distribution of MDEs. Briefly, PBS-suspended MDEs were loaded into the sample chamber of the NanoSight unit, a laser source at 532 nm was applied to the diluted MDE suspension, and a video was recorded for 60 s at a frame rate of 24.98 fps. The movement of particles was analyzed using NTA software.

### 2.4. Dynamic Light Scattering

We performed dynamic light scattering (DLS) and zeta potential determinations using a Zetasizer nanoseries instrument (λ = 532 nm laser wavelength) (Malvern Nano-Zetasizer, Worcestershire, UK). The MDE size data referred to the distribution of scattering intensity (z average).

### 2.5. HSCs’ Isolation and Cell Culture

HSCs were isolated from BALB/c mice livers, using techniques of enzymatic dissociation and gradient separation. This was performed by using pronase–collagenase perfusion, followed by Histodenz density gradient centrifugation, as described by Friedman et al. [21]. This method is well established in our lab. HSCs were then cultured for varying periods of time, and incubated with MDEs, depending on the experiment (Appendix A).

### 2.6. MDEs’ Feeding by Gavage

To assess the effects of MDEs on HSCs in vivo, BALB/c mice received 13 mg/kg cow MDEs in 200 μL PBS, orally by gavage, for 10 days, in the experiments described in Appendix A and for 24 h in the MDE-uptake experiments. The control group received 200 μL of PBS, orally by gavage, for 10 days. On the 10th day, the mice were sacrificed, and their liver was removed and used for HSCs’ isolation (Appendix A).

### 2.7. Exosomes Labeling

For in vitro usage: MDEs were labeled using the Exo-Glow Exosome Labeling Kit (System Biosciences, San Francisco, CA, USA). We added 20 μL of 10× Exo-Red to 200 μL of a resuspended-exosome suspension in PBS. The mixture was mixed well by inversion and incubated for 10 min at 37 °C. To stop the labeling reaction, we added 40 μL of the ExoQuick-TC reagent to the labeled exosome sample suspension and mixed it by inverting six times. The labeled EV samples were incubated on ice for 30 min. Samples were then centrifuged at 12,000× *g* for 3 min, to sediment the exosomes, and the pellet was resuspended in 200 μL of PBS.

For in vivo usage: MDEs were incubated with 1 µM fluorescent lipophilic tracer 1,1-dioctadecyl-3,3,3,3-tetramethylindotricarbocyanine iodide (DiR) (INVITROGEN, Carlsbad, CA, USA) at 37 °C for 15 min. Following incubation, PBS was added to the mix, which was then centrifuged at 100,000× *g* for 60 min. Labeled EVs were pelleted, whereas any unbound label was discarded.

### 2.8. Cell Proliferation Assessment

HSCs’ proliferation was measured using methylene blue assay and 3-(4,5-dimethylthiazol-2-yl)-2,5-diphenyl-2H-tetrazolium bromide (MTT) assay.

Methylene blue assay: For the methylene blue assay, cells were fixed in glutaraldehyde at a final concentration of 0.05% for 10 min at room temperature. After washing, the cells were stained with 1% methylene blue in 0.1 M borate buffer, pH 8.5, for 60 min at room temperature. The cells were then washed extensively and rigorously, to remove excess dye, and then dried. The dye taken up by cells was eluted in 0.1 M HCl for 60 min at 37 °C, and absorbance was monitored at 620 nm. The cells were photographed after fixation and methylene blue staining.

MTT assay: The MTT assay was carried out using the CellTiter96 nonradioactive cell proliferation kit (Promega Corp, Madison, WI, USA) and MTT-based assay, according to the manufacturer’s recommendations. After isolation, cells were seeded in a 96-well plate. At the end of the treatment period, the medium was changed, and cells were incubated with the dye solution for 4 h at 37 °C, followed by solubilization in dimethyl sulfoxide, and spectrophotometric measurement was undertaken. The rate of formazan dye formation was determined by measuring the absorbance (570 nm–640 nm). The 570 nm–640 nm reading value is directly proportional to the number of living cells.

### 2.9. Immunoblotting

After being cultured, cells were lysed in SDS, separated by SDS-PAGE, and transferred onto a PVDF membrane. The membranes were probed with primary antibodies against collagen Iα1 (Abcam, Cambridge, MA, USA), PPARγ (Abcam, Cambridge, MA, USA), β-actin (Abcam, Cambridge, MA, USA), CD9 (1:1000; SBI System Biosciences, Palo Alto, CA, USA), CD81 (1:1000; Cosmo Bio, Tokyo, Japan) and β-casein (Abcam, Cambridge, MA, USA). β-actin was used as a marker for protein load control. The secondary antibody was horseradish per-oxidase (HRP)-conjugated goat, anti-mouse or anti-rabbit (1:3000; Cell Signaling Technology, Danvers, MA, USA).

### 2.10. RNA Extraction

From EVs: Trizol reagent (INVITROGEN, Carlsbad, CA, USA) was added to the MDE pellet [19]. Extraction of RNA was performed as previously described [19].

From cell culture: Cells were collected and suspended with TRIzol reagent (Invitrogen, Paisley, UK), for further extraction of RNA. The isolation of total RNA was undertaken using the Zymo Direct-zol RNA MiniPrep Kit (Zymo Research, Irvine, CA, USA), according to the manufacturer’s protocols. RNA quantity and quality were assessed by measuring the absorbance at different wavelengths, using a NanoDrop spectrophotometer (Waltham, MA, USA), in the RNA samples.

### 2.11. mRNA Detection by qRT-PCR

For the quantification of mRNA, complementary cDNA was generated using the high capacity RNA-cDNA kit (Applied Biosystems, Foster City, CA, USA), according to the manufacturer’s instructions. A quantity of 1 µg total RNA isolated HSCs was used to generate cDNA. The mRNA levels were measured using qRT-PCR with master mix (Fast qPCR SyGreen Blue Mix, PCR Biosynthesis, Wayne, PA, USA). The PCR reaction steps were performed as previously described [19]. The 2^(−ΔΔCT) method was used to determine the relative amounts of mRNA, using GAPDH as the reference gene. For primers list, see online Appendix A).

### 2.12. MicroRNA Detection by qRT-PCR

500 ng of total RNA extracted from HSCs and 100 ng of RNA extracted from MDEs were used to prepare cDNA. The qScript micro-RNA cDNA Synthesis Kit (Quantabio, Beverly, MA, USA) was used to reverse-transcribe the RNA to cDNA, according to the manufacturer’s instructions, and the resulting cDNA was used to assess the expression of miR-148a, miR-320, miR-21, Let-7a, miR-29 and RNU6. RNU6 was used as a reference for normalization of miRNA expression levels. The PerfeCTa SYBR Green SuperMix (Quantabio, Beverly, MA, USA) was used, together with Quantabio micro-RNA qPCR primers, for the miR-148a-3p (HSMIR-0148A-3P), miR-320a (HSMIR-0320A), miR-375 (HSLET-0007A-5P), Let-7a (HSLET-0007A-5P) and RNU6 (HS-RNU6), obtained from Quantabio (Beverly, MA, USA). The qRT-PCR was run using a two-step cycling protocol, as previously described [17]. Normalization and relative expression level calculation was carried out using the 2^(−ΔΔCT) method.

### 2.13. Ethical Approval Information

This study was approved by the Investigational Review Board (IRB) of Hadassah-Hebrew University Hospital (HM0-0101-13). All the mothers who donated milk in this study signed informed consent forms.

The protocols of this study, in the DSS-induced colitis model in mice, were approved by the Ethics Committee—research number: MD-20-15923-4.

## 3. Results

### 3.1. Isolation and Characterization of Milk-Derived Extracellular Vesicles

Extracellular vesicles (EVs), isolated from cows’- and human milk, were characterized by size protein and miRNA expression. Milk-derived EVs’ (MDEs’) size was determined by nanoparticle tracking analysis (NTA) and dynamic light scattering (DLS). The mean size of the MDEs was 156 nm (Figure 1A). The Zeta-average (d-nm) of the MDEs was 186.1 nm, as shown in Figure 1B. The polydispersity index (PDI) of the MDE was 0.208 (Figure 1B), revealing a relatively even size distribution of the MDEs which could also be confirmed by the sharp single main peak in our NTA analysis. The exosome-related protein expression on the MDEs was assessed by western blot analysis. The MDEs expressed CD9 and CD81, two of the main exosome-related proteins (Figure 1C). The purity of the MDEs was determined by western blot analysis. MDEs expressed the CD9 and CD81 exosome-related proteins, whereas β-casein, a non-exosomal protein, was only detected in the fat and casein fraction of the milk, indicating that the isolated MDEs were highly purified and not contaminated by other components, such as casein micelles (Appendix A). MiRNAs are one of the main cargos of EVs, including exosomes. Indeed, we analyzed the expression of several miRNAs from the MDEs, as shown in Figure 1D. Taken together, these results demonstrated that extracellular vesicles, mainly exosomes, were successfully isolated from milk, with high purity, and were further efficiently characterized using various methods.

### 3.2. Uptake of MDEs In Vivo and In Vitro

Isolated labeled MDEs from cows’ milk were incubated with HSCs. The cells were visualized by bright-field illumination and fluorescence microscopy. Following incubation with MDEs, HSCs were positively stained, which indicates uptake of MDEs into those cells (Figure 2A).

MDEs isolated from cows’ milk and human milk were labeled with an infrared fluorescent membrane dye, namely DiR dye, to track their localization patterns in vivo. Following 24 h of the gavage administration of labeled cow- or human MDEs, imaging revealed an accumulation of the fluorescent signal in the liver (Figure 2B).

### 3.3. MDEs Inhibit HSCs’ Proliferation

To examine whether MDEs have an effect on the proliferation of HSCs, we measured proliferation rates in HSCs incubated with MDEs, at days 4, 7 and 14, following HSC isolation (during the in vitro activation of the HSC by plate growth) (Appendix A). MDEs isolated from cows’ or human milk reduced the growth rate of HSCs, compared with those HSCs grown without MDEs (Figure 3). HSCs were incubated with 0.1 mg/mL and 0.01 mg/mL MDEs. Higher MDEs’ concentration increased the inhibition effect on HSCs proliferation, in comparison with the lower concentration (Figure 3). Furthermore, cell proliferation was measured, following stimulation with platelet-derived growth factor (PDGF), MDEs, or a combination of the two, for 24 h (Appendix A). PDGF increased HSC proliferation ratio by ~1.4 fold, while the combined treatment of PDGF and MDEs decreased HSC proliferation to the control level (ratio of ~0.5 suppression). Interestingly, treatment with MDEs alone decreased cell proliferation by a ratio of ∼0.6 compared to control levels, suggesting that treatment with MDEs suppresses proliferation of both PDGF-stimulated- and unstimulated HSCs (Figure 4A).

### 3.4. Downregulation of Collagen Expression and Upregulation of Peroxisome Proliferator-Activated Receptor—γ (PPAR-γ) by MDEs

We analyzed the protein expression levels of collagen type I in primary HSCs, following incubation with TGFβ, MDEs, or a combination of the two, after 24 h (Appendix A). TGFβ treatment increased the expression of collagen I compared to the control treatment, while the incubation with MDE of TGFβ-treated cells led to inhibition of collagen I (Figure 4B). Incubation of MDEs during HSC treatment have no effect on collagen expression (Appendix A). In contrast, TGFβ treatment decreased the expression of PPAR-γ compared to the control treatment, while the incubation with MDE of TGFβ-treated cells led to upregulation of PPAR-γ expression (Figure 4C).

### 3.5. Regulation of miRNA Expression by MDEs

We analyzed the expression of miR-148, one of the highly expressed miRNAs, in MDEs in HSCs, following incubation with TGFβ, or TGFβ and MDEs (Appendix A). Incubation with MDE and TGFβ-treated cells led to higher expression of miR-148a. The treatment of HSC with TGFβ had no effect on miR-148a expression level (Figure 4D).

### 3.6. The Effect of MDEs on HSCs In Vivo

To examine the effect of MDEs on HSCs in vivo, mRNA expression and miRNA content in HSCs isolated from livers of mice that were treated with MDEs (Appendix A) was evaluated. We analyzed the gene expression of α-SMA, collagen, and TIMP1. The expression of α-SMA, collagen, and TIMP1 was downregulated in HSCs isolated from MDE-treated mice (Figure 5A). The expression of miR-148 and let-7 was shown to be higher in HSCs isolated from MDE-treated mice. In contrast, the expression of miR-21 was noted to be lower in HSCs isolated from MDE- treated mice. The expression of miR-29 was not regulated in HSCs following MDE treatment (Figure 5B).

## 4. Discussion

Extracellular vesicles such as exosomes are nanoparticles containing a wide variety of proteins, lipids, mRNA and miRNA. Extracellular vesicles can be found in most human biological fluids, and breast milk is known to contain the highest content [22]. MDEs are bioactive products that enter into target cells and transfer their cargo, such as miRNAs, into cells, and induce biologic change in them [17,19,23]. MiRNAs have been linked to fibrosis in various organs and disease settings. Most of these miRNAs directly induce or protect from fibrosis by targeting TGF-β pathways, connective tissue growth factor (CTGF), and ECM structural proteins. Among the many miRNAs known to be involved in fibrosis, there are miRNAs that have a beneficial effect on fibrosis, such as the let-7 family, miRNA-107, miRNA-148, miRNA-29 and miRNA-30 [24,25,26,27]. However, there are also miRNAs that are known to have a pro-fibrotic effect, such as miRN-142, miRNA-192, and miRNA-21 [28,29,30].

We have shown that MDEs are able to enter into HSCs in vitro and into the liver in vivo (Figure 2) and, subsequently, inhibit HSC proliferation (Figure 3 and Figure 4B). Moreover, treatment with MDEs, in vivo and in vitro, results in an increase of miRNA-148 expression in HSCs. MiRNA-148 was found to inhibit the proliferation of tumor cells in tumors such as breast cancer, gastric cancer, urogenital cancer and liver cancer [31,32,33,34]. There are also studies that have shown that overexpression of miRNA-148 resulted in apoptotic and autophagic activity of HSCs, whereas the knocking-down of miRNA-148 contributed to cell proliferation [12]. Based on these findings, it is plausible that the inhibition of the proliferation of HSCs by MDEs may be due to the increased expression of miR-148.

In this study, we found that MDE treatment of mice led to a decrease in collagen type 1 expression and an increase in PPAR-γ expression in HSCs (Figure 4A,B). Type I collagen is a marker of HSCs’ activation and the main ECM protein in liver fibrosis; PPAR-γ, however, is a gene in which expression and activity are suppressed in activated HSCs [35]. Moreover, an mRNA analysis for HSCs isolated from the livers of mice treated with MDEs revealed that the expression of α-SMA, collagen, and TIMP1 genes, was downregulated, due to MDEs treatment (Figure 5). α-SMA, type I collagen and TIMP1 are fibrogenic markers and their presence indicates biotransformation of quiescent HSC to myofibroblast cells, with secretion of ECM proteins [36]. In addition, miRNA-148 has an inhibitory effect on the activation of HSCs, by the regulation of transforming growth factor β receptor type 1 (TGFBR1) and thus the blocking of its recruitment by TGF-β [37]. These results have led us to conclude that MDEs inhibit, in part, the activation of HSCs. We hypothesize that much of the effect we have seen of MDEs on HSCs’ proliferation and activation may be due, in part, to the presence of miRNA-148 within MDEs and the increase in its level within HSCs, following treatment with MDEs. It is quite possible that the effect of miRNA-148 on HSCs is due to its interaction with targets investigated in other studies, such as TGFBR1 and ERBB3 [25,37].

Let-7 was also, like miRNA-148, one of the most common miRNAs in the MDEs, and we found that similar to miRNA-148, its level also increased in HSCs isolated from the livers of mice treated with MDEs. MiRNAs from the let-7 family have been found to be associated with the spreading and invasiveness of HCC, and their expression has been found to be associated with the spread of the tumor and its resistance to chemotherapy [38]. Studies have also found that let-7 levels decrease in the livers of mice with alcoholic liver disease (ALD), and that administration of ethanol in vitro and in vivo lowered let-7 levels and increased fibrosis markers. In addition, the knock-out of let-7 in HSCs led to an increase in their activation, as evidenced by a decrease in the activation markers of HSCs such as collagen, α-SMA and TIMP [24]. Proliferation and activation of HSCs are important conditions for the development of liver fibrosis, and the results we show in this study suggest that treatment with MDEs inhibits these conditions. Based on what we see in the literature regarding let-7, we can conclude that its high presence in the MDEs with which we treated HSCs is part of the reason why we observed inhibition of proliferation and activation, following treatment.

Another common miRNA we found in the exosomes isolated from milk was miRNA-21, but unlike miRNA-148 and let-7, we found that in MDE-treated HSCs the level of miRNA-21 decreased. MiRNA-21 was found in high levels in the livers of mice that had undergone arsenic-induced liver fibrosis. Knock-out of miRNA-21 in mice exposed to arsenate attenuated hepatic fibrosis [39]. In addition, treatment of HSCs with PDGF increased synthesis and expression of type 1 collagen and α-SMA, which are markers for the activation of HSCs, and simultaneously increased expression of miRNA-21. Moreover, downregulation of MIR-21 expression in HSCs prevented PDGF-induced cell activation, and upregulation of miR-21 alone stimulated the activation of HSCs [40]. The decreased expression of the pro-fibrotic miR-21 in HSC following MDE treatment may contribute to inhibition of HSC activation by MDEs.

Extracellular vesicles, in particular, exosomes, have attracted significant attention as a therapeutic tool in the last few years. In some studies, EVs have been used as drug-delivery vehicles in several conditions, such as: inflammation, cancer, and Parkinson’s disease [41,42,43]. In addition, studies have examined the effect of EVs as regulators of cell activity and even as a therapeutic tool, based on the load they carry and depending on the tissue from which they were secreted [44]. Studies have even found that treatment with EVs of different origins may improve liver damage and fibrosis [5]. For example, miR-181-5p-modified murine exosomes secreted by adipose-derived mesenchymal stem cells (ADSCs) have an anti-fibrotic effect and amelioration of liver function [45]. Another study showed that EVs isolated from human-induced pluripotent stem cells modulate HSCs’ activation and may have an anti-fibrotic effect [46]. MDEs have protective effects on cardiac fibrosis, alleviate the extracellular matrix deposition and enhance the cardiac function. Even though, as far as we know, no studies have examined the effect of MDEs on liver fibrosis. In this work, we have shown for the first time that MDEs can enter HSCs and we have shown that MDEs have potential therapeutic modality on liver fibrosis. MDEs have recently emerged as a potential new treatment modality for autoimmune and inflammatory diseases, such as colitis [20], rheumatoid arthritis [47] and cardiac fibrosis [48]. We hypothesize that the mechanism by which MDEs may constitute a therapeutic tool in liver fibrosis is related to their miRNA cargo and the way in which they affect the proliferation and activation of HSCs.

## 5. Conclusions

In conclusion, MDE inhibit the proliferation and fibrogenic mediators expression such as collagen, α-SMA and TIMP1 and upregulate PPAR-γ expression in HSCs. MDEs alter the composition of the miRNA profile within HSCs and, presumably by this means, affect their proliferation and activation. These results suggest that MDEs have an anti-fibrotic effect on the liver and may be a part of future treatment for liver fibrosis.

## Figures and Tables

**Figure 1 nutrients-14-04049-f001:**
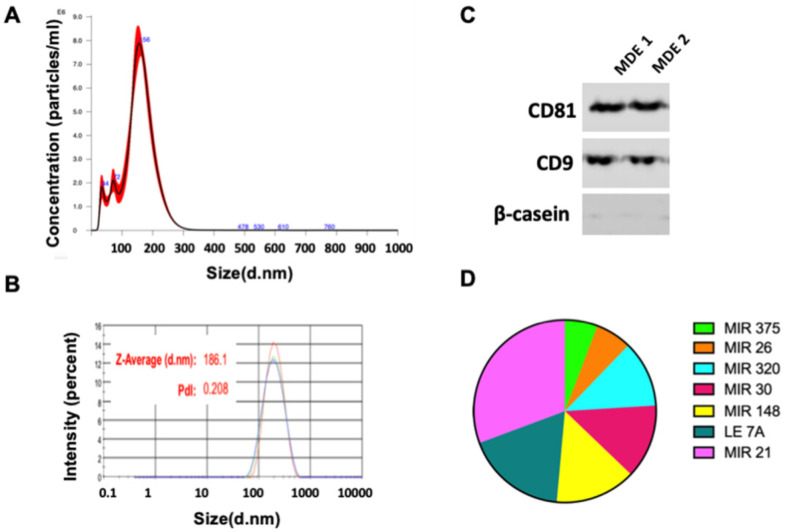
Isolation of extracellular vesicles from milk. Milk derived extracellular vesicles (MDEs) were isolated from cows’ milk by sequential centrifugation. The profile of the size distribution of MDEs was investigated using NanoSight. (**A**) The distribution of the particle size (by intensity) of isolated MDEs was determined by dynamic light scattering (DLS) (**B**). The protein expression of the Cluster of Differentiation 81 (CD81) and CD9 exosome-specific markers and β-catenin was evaluated via western blotting (**C**). The expression of abundant microRNA (miRNA) in MDEs was measured by qRT-PCR. The results of qRT-PCR analysis are shown as Ct values (**D**). Pdl = Polydispersity Index.

**Figure 2 nutrients-14-04049-f002:**
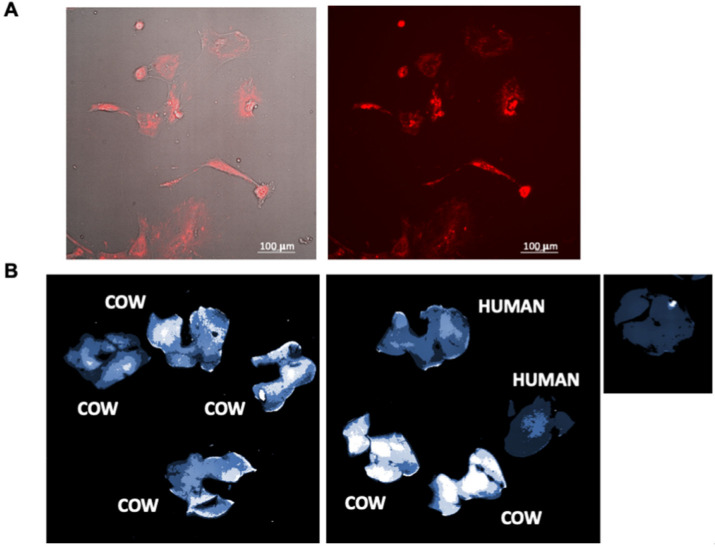
MDEs are absorbed by liver cells and HSCs. Labeled cow MDEs (0.1 mg/mL) were incubated for hours with HSCs. Bright-field images in combination with fluorescent images, and fluorescent images, were obtained by fluorescence microscope analysis (**A**). Each Balb/c mouse was administered by gavage, DiR dye-labeled cow- or human MDEs (in 200 μL Phosphate-buffered saline (PBS)). Control mice receive PBS by gavage. Mice were sacrificed following 24 h of MDE gavage, and images were obtained for fluorescent analysis using the Typhoon FLA 9500 scanner (**B**), *n* = 4.

**Figure 3 nutrients-14-04049-f003:**
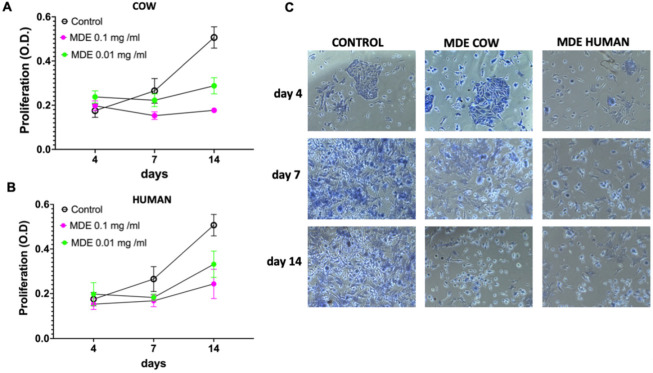
Proliferation of HSCs incubated with MDEs. Cell proliferation of HSCs grown without MDEs (Control) and with MDEs from cow (COW MDE) and human milk and (HUMAN MDE), in different concentrations, was examined by methylene blue staining (**A**,**B**). Light microscopy pictures of selected fields following MDE treatment during the different times of incubation (**C**).

**Figure 4 nutrients-14-04049-f004:**
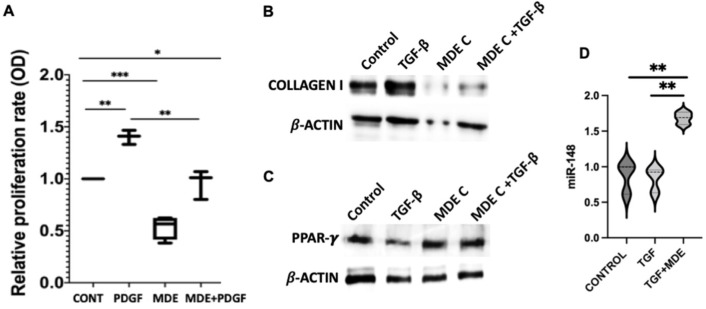
Proliferation, protein expression and miRNA expression in TGF-β and PDGF activated HSCs incubated with MDEs. Cell proliferation of HSCs, activated by PDGF and treated with MDEs, was examined by MTT analysis. HSCs were cultured in 0% FCS and activated by PDGF. HSCs incubated with cow (MDE) or without cow MDEs (CONT) and HSCs incubated with PDGF with (PDGF) or without PDGF and cow MDEs (MDE + PDGF), error bars represent SEM (**A**) (*n* = 4). Expression of collagen type I (**B**) and PPAR-γ (**C**) proteins was analyzed in untreated HSCs (Control) and in TGF-β-activated HSCs (TGF-β) compared to HSCs incubated with cow MDEs (MDE) and TGF-β-activated HSCs incubated with cow MDEs (MDE + TGF-β). Expression of miRNA-148 in TGF-β-activated HSCs (TGF-β) compared to HSCs incubated with cow MDEs (TGF-β + MDE) and untreated HSCs (CONTROL). Expression of miRNAs was analyzed by qRT-PCR. Values were calculated using the 2^(−ΔΔCT) method and normalized against the expression of RNU6 (**D**). * *p* < 0.05, ** *p* < 0.01, *** *p* < 0.001.

**Figure 5 nutrients-14-04049-f005:**
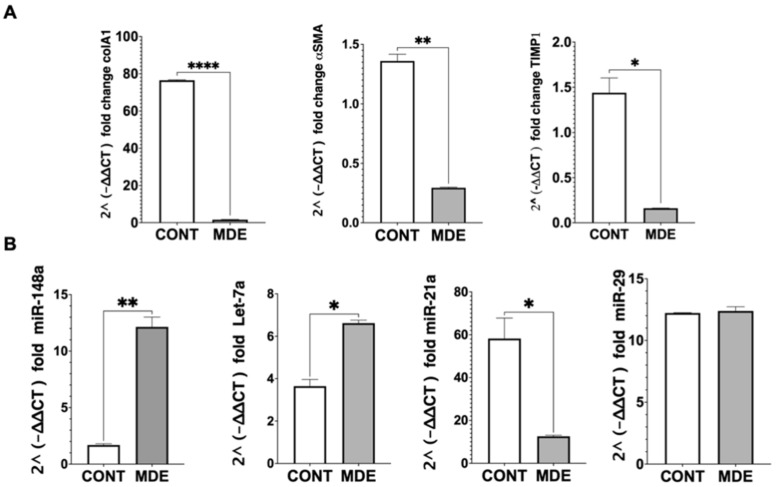
Expression of gene and milk abundant miRNA in HSCs isolated from cow MDE-treated mice. Expression of α-SMA, collagen, and TIMP1 genes in HSCs isolated from MDE-treated (MDE) and untreated (CONT) mice. The expression level of those genes was analyzed by qRT-PCR. Values were calculated using the 2^(−ΔΔCT) method, and normalized against the expression of GAPDH (**A**). Expression of miRNA-148, miRNA-29, let-7, and miRNA-21 in HSCs isolated from MDE-treated (MDE) and untreated (CONT) mice. Expression of miRNAs was analyzed by qRT-PCR. Values were calculated using the 2^(−ΔΔCT) method and normalized against the expression of RNU6 (**B**). Data are mean ± SEM. * *p* < 0.05, ** *p* < 0.01, **** *p* < 0.0001.

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
