# Peer review of "Beneficial Effects of Milk-Derived Extracellular Vesicles on Liver Fibrosis Progression by Inhibiting Hepatic Stellate Cell Activation"

_nutrients, 2022, doi:10.3390/nu14194049_

Round 1
Reviewer 1 Report
Authors evaluated whether MDEs regulate fibrotic progression by inhibiting HSCs activation via miRNA content. To study this question, they study the effect of human and cow MDE on HSCs isolated from murine livers. They show that MDEs regulate proliferation and activation of HSC. Study is novel and the results of this study might provide a new approach for the treatment of liver fibrosis.
Major :
Methodology needs to improve – Author needs to disclose the sources of human EV and consents received from donors; it is not clear from the protocol, what they followed with human MDEs? (Whether it is given for the mouse or whether it for cell treatments?)
Figure 2 A- Author needs to give better description about the images, also clarify the results in between two figures.
Figure 2 B- Author needs to give clear information about, why cow and human fluorescence is in same picture, whether they administered the MDEs from cow and human both together?
Figure 2- Author shows fluorescence images of MDE absorbed by liver and HSE cells, but author needs to show these images with colocalization with HSC markers (Desmin) to show that MDE intake into HSCs
Figure 3- Needed qPCR data for the proliferation markers (ex: PCNA, Ki67) with different concentrations of MDE with cow and human. Methylene blue assay pictures can vary from one field to another, therefore not reliable
Figure 4 – Author needs to show the activation of HSC with PDGF, and how it affects with collagen and fibrosis levels.
Rationale of choosing miRNA-148 expression in HSC is not clear, author checks the Let-7 expression and discuss that high presence in the MDEs maybe the reason why they see inhibition of proliferation and activation following treatment. There are multiple other regulators of this expression (ex: Let-7- Lin 28 works in see saw fashion, and Lin 28 inhibit the precursors of let-7 (PMID: 27012771)) Author needs to check levels of Lin28 in HSCs
Minor :
Used lot of abbreviations, some are undefined (ex: PDGF, DNMTs)
Author Response
Authors evaluated whether MDEs regulate fibrotic progression by inhibiting HSCs activation via miRNA content. To study this question, they study the effect of human and cow MDE on HSCs isolated from murine livers. They show that MDEs regulate proliferation and activation of HSC. Study is novel and the results of this study might provide a new approach for the treatment of liver fibrosis.
Major:
- Methodology needs to improve – Author needs to disclose the sources of human EV and consents received from donors; it is not clear from the protocol, what they followed with human MDEs? (Whether it is given for the mouse or whether it for cell treatments?)
Thank you for this comment according to the reviewer recommendation we added two new subsections in the materials and methods section: Milk sample collection and Ethical Approval Information.
We made it clearer in the revised version about the source of MDEs. We added this information in the figure legends.
- Figure 2 A- Author needs to give better description about the images, also clarify the results in between two figures.
We add more explanation of the two pictures in 2A.
- Figure 2 B- Author needs to give clear information about, why cow and human fluorescence is in same picture, whether they administered the MDEs from cow and human both together?
Figure 2B left shows the results following gavage administration of MDE isolated from COW given to four different mice. The four liver of those mice are show in the left picture. In the right picture we show the liver of two mice following gavage administration with cow MDE and another two liver from mice that receive human MDE. Each mouse received by gavage either human or cow MDEs. We clarify it the revised manuscript.
- Figure 2- Author shows fluorescence images of MDE absorbed by liver and HSE cells, but author needs to show these images with colocalization with HSC markers (Desmin) to show that MDE intake into HSCs
HSC were isolated using a well-accepted and published method (Ref #21). We published as others several papers with this methodology [1]. Moreover, the shape of the cells is typical to HSC. Moreover, their protein expression of collagen and alpha SMA are unique to HSC.
In the figure 2B that shows the absorption by the liver, our aim was to show that MDE are absorbed by the liver.
- Figure 3- Needed qPCR data for the proliferation markers (ex: PCNA, Ki67) with different concentrations of MDE with cow and human. Methylene blue assay pictures can vary from one field to another, therefore not reliable
Indeed, the reviewer has a right concern. To clarify this point: each picture represents at least four different experiments. Moreover, the methylene blue assay (Figure 3A) quantify the effect on all cells in each experiment. The propose of pictures in Figure 3B is to show the effect of MDEs treatment on HSC cells during activation time.
- Figure 4 – Author needs to show the activation of HSC with PDGF, and how it affects with collagen and fibrosis levels.
PDGF effect is mainly on HSC proliferation therefore we used PDGF to induce proliferation. On the other hand, TGF-β is profibrotic mediator and we use TGF-β to measure the effect on fibrotic parameters such as collagen expression. However according to the reviewer recommendation, we analyze the effect of MDE on PDGF activated cells. No significant effect was detected as expected and the results were now added to the supplementary data.
- Rationale of choosing miRNA-148 expression in HSC is not clear, author checks the Let-7 expression and discuss that high presence in the MDEs maybe the reason why they see inhibition of proliferation and activation following treatment. There are multiple other regulators of this expression (ex: Let-7- Lin 28 works in see saw fashion, and Lin 28 inhibit the precursors of let-7 (PMID: 27012771)) Author needs to check levels of Lin28 in HSCs
We choose to analyze the expression of miR-148 based on our and other studies that miR-148 (Ref #17, [2]) is a dominant miRNA in milk and is known as a miRNA with anti-fibrotic function (Ref #25). Moreover, previous studies proved its role on cell proliferation and differentiation (Ref #19).
As suggested by the reviewer we analyzed the level of Lin28 in HSC following MDE treatment and no differences were observed as shown below.
Minor :
Used lot of abbreviations, some are undefined (ex: PDGF, DNMTs)
We defined all the abbreviation on the revised manuscript.
[1] Abramovitch S, Dahan-Bachar L, Sharvit E, Weisman Y, Ben Tov A, Brazowski E, et al. Vitamin D inhibits proliferation and profibrotic marker expression in hepatic stellate cells and decreases thioacetamide-induced liver fibrosis in rats. Gut 2011;60:1728–37. https://doi.org/10.1136/gut.2010.234666.
[2] Li W, Li W, Wang X, Zhang H, Wang L, Gao T. Comparison of miRNA profiles in milk-derived extracellular vesicles and bovine mammary glands. Int Dairy J 2022;134:105444. https://doi.org/10.1016/j.idairyj.2022.105444.
Reviewer 2 Report
The paper of Reif et al entitled ”Beneficial effects of milk-derived extracellular vesicles on liver fibrosis progression by inhibiting hepatic stellate cell activation” presents interesting and well designed experiments regarding the inhibition of liver fibrosis due to milk-derived extracellular vesicles. The novelty of study is great. However, the drafting is sloppy in some places. So, in the line 324 of page 9, the word starting with k is missing. Also, the lines 370-373 from page 12 are probably recommendations for drafting and have to be removed. Also the conclusions have to be improved in order to be more precise.
Author Response
The paper of Reif et al entitled ”Beneficial effects of milk-derived extracellular vesicles on liver fibrosis progression by inhibiting hepatic stellate cell activation” presents interesting and well designed experiments regarding the inhibition of liver fibrosis due to milk-derived extracellular vesicles. The novelty of study is great. However, the drafting is sloppy in some places. So, in the line 324 of page 9, the word starting with k is missing. Also, the lines 370-373 from page 12 are probably recommendations for drafting and have to be removed. Also the conclusions have to be improved in order to be more precise.
We corrected the style of the paper according to the reviewer request.
As suggested by the reviewer we improved the conclusions in the revised manuscript.
